# Artificial Intelligence in Endoscopic Ultrasonography-Guided Fine-Needle Aspiration/Biopsy (EUS-FNA/B) for Solid Pancreatic Lesions: Opportunities and Challenges

**DOI:** 10.3390/diagnostics13193054

**Published:** 2023-09-26

**Authors:** Xianzheng Qin, Taojing Ran, Yifei Chen, Yao Zhang, Dong Wang, Chunhua Zhou, Duowu Zou

**Affiliations:** Department of Gastroenterology, Ruijin Hospital, School of Medicine, Shanghai Jiao Tong University, Shanghai 200025, China; qin_xz@sjtu.edu.cn (X.Q.); rantaojing@163.com (T.R.); a244715494@163.com (Y.C.); zyrjxh97@sjtu.edu.cn (Y.Z.); drdongwangrj@163.com (D.W.)

**Keywords:** artificial intelligence, solid pancreatic lesions, computer-aided diagnosis, endoscopic ultrasound-guided fine-needle aspiration, endoscopic ultrasound-guided fine-needle biopsy

## Abstract

Solid pancreatic lesions (SPLs) encompass a variety of benign and malignant diseases and accurate diagnosis is crucial for guiding appropriate treatment decisions. Endoscopic ultrasonography-guided fine-needle aspiration/biopsy (EUS-FNA/B) serves as a front-line diagnostic tool for pancreatic mass lesions and is widely used in clinical practice. Artificial intelligence (AI) is a mathematical technique that automates the learning and recognition of data patterns. Its strong self-learning ability and unbiased nature have led to its gradual adoption in the medical field. In this paper, we describe the fundamentals of AI and provide a summary of reports on AI in EUS-FNA/B to help endoscopists understand and realize its potential in improving pathological diagnosis and guiding targeted EUS-FNA/B. However, AI models have limitations and shortages that need to be addressed before clinical use. Furthermore, as most AI studies are retrospective, large-scale prospective clinical trials are necessary to evaluate their clinical usefulness accurately. Although AI in EUS-FNA/B is still in its infancy, the constant input of clinical data and the advancements in computer technology are expected to make computer-aided diagnosis and treatment more feasible.

## 1. Introduction

Solid pancreatic lesions (SPLs) comprise a variety of both benign and malignant diseases, including pancreatic ductal adenocarcinoma (PDAC), pancreatic neuroendocrine tumors (PNET), focal pancreatitis, pancreatic tuberculosis, and pancreatic metastasis [1]. Accurate diagnosis plays a crucial role in guiding appropriate treatment decisions. The clinical presentation of solid pancreatic lesions is highly variable and primarily depends on the histological pattern, location, and size of the lesion. Typically, a pancreatic head tumor results in reduced exocrine function or biliary duct obstruction, both of which result in jaundice [2]. In contrast, a mass in the pancreas’ body and tail is frequently asymptomatic [3]. PDAC is one of the most frequent types of pancreatic lesions and the fifth leading cause of cancer death worldwide, with an overall 5-year survival rate of less than 5% [4]. Only around 10% of patients qualify for potentially curative surgery at diagnosis, and survival gains have been negligible, with patients commonly suffering from illness recurrence [5]. Suppose the SPL is a PNET, particularly a functional one. In that case, its symptoms are related to its hormone-producing capabilities, such as insulin, gastrin, vasoactive intestinal peptide, glucagon, somatostatin, and serotonin, which make it easier to identify [6]. Therapeutic choices are mostly reliant on the capacity to identify or rule out malignancy because the clinical course and prognosis of pancreatic masses vary. Therefore, establishing an optimal treatment strategy depends heavily on the correct diagnosis. Endoscopic ultrasound (EUS) is a well-established tool for evaluating pancreatic lesions. EUS offers high spatial resolution observation of the pancreas due to its proximity; the sensitivity of pancreatic cancer identification has reached 94% [7,8]. Endoscopists can also perform a fine-needle aspiration/biopsy (FNA/B) of tumors under EUS guidance to obtain cytological and histological diagnoses. Additionally, contrast-enhanced harmonic EUS (CH-EUS) and EUS elastography provide complementary information to conventional EUS in diagnosing pancreatic lesions, leading to more accurate diagnoses [8]. The use of contrast agents in EUS can provide valuable information on the microvasculature and perfusion within organs of interest and hypoenhancing masses have been proven to be indicators of malignant tumors [9]. According to a range of studies previously reported, CH-EUS shows a sensitivity range of 80% to 96% and a specificity range of 64% to 100% for SPLs, particularly in differentiating PDACs from other pancreatic masses [10]. Utilizing CH-EUS as an adjunct to assist EUS-FNA demonstrates higher diagnostic sensitivity in the assessment of pancreatic masses compared to standard EUS-FNA (CH-EUS-FNA, 84.6%; EUS-FNA, 75.3%) [11]. As for EUS elastography, it serves as a valuable complement to tissue sampling, which is used to guide fine-needle punctures and aid in determining further clinical treatment plans [12]. Elastography has been reported to demonstrate an extremely high sensitivity (ranging from 92% to 98%) in the detection of malignant pancreatic tumors [13]. Specifically, it has an exceptionally high negative predictive value for diagnosing PDAC in small pancreatic lesions [14].

EUS-FNA has been considered a safe and accurate procedure for diagnosing pancreatic lesions since the study reported by Vilmann et al. in 1992 [15]. According to the guidelines provided by the European Society of Gastrointestinal Endoscopy (ESGE) and the National Comprehensive Cancer Network (NCCN), SPL patients without a definitive diagnosis need to undergo EUS-FNA for pathological diagnosis [16,17]. Furthermore, EUS-FNA is recommended for pathological diagnosis before neoadjuvant therapy is administered and for patients with locally advanced, unresectable pancreatic cancer or metastatic disease [17]. FNA’s reported sensitivity and specificity for PC were 85–92% and 96–98%, respectively [18,19]. Compared to EUS-FNA, EUS-FNB can collect a larger amount of tissue and preserve the associated architecture of the surrounding tissue, which aids in the definitive diagnosis of suspicious pancreatic lesions. Recent studies have found that EUS-FNB has better diagnostic accuracy than EUS-FNA for suspicious pancreatic lesions and requires fewer needle passes, leading to a shorter time to diagnosis [20,21]. However, these findings do not standardize the used needle sizes and locations of the lesions. After settling on a 22G needle size, the diagnostic accuracy of FNB has not been proven to be significantly better than FNA [22]. While a network meta-analysis suggests that Franseen and fork-tip needles may offer superior performance in the tissue sampling of pancreatic masses, the confidence level in these estimates remains low [23]. Therefore, EUS-FNA/B serves as a first-line minimally invasive diagnostic tool for pancreatic mass lesions and is widely used in clinical practice, especially when the diagnosis or staging of the disease is unclear or when neoadjuvant therapy is planned.

In the past decade, an increasing number of medical centers have been performing EUS-FNA/B to improve diagnostic accuracy for pancreatic masses. EUS-FNA/B is a multistep procedure and its accuracy is influenced by various uncertain factors, including rapid on-site evaluation (ROSE) with cytopathologist involvement [24]. ROSE is a technique where experienced cytopathologists assess the quality of aspirate or biopsy samples on-site through the rapid creation of smears and staining methods. ROSE, in EUS-guided tissue acquisition, is used to assess sample adequacy and the nature of lesions in real-time, which can enhance diagnostic accuracy, reduce needle passes, and decrease the proportion of inadequate samples [25,26]. However, not all EUS centers have cytopathologist staffing available and, even when they are present, the ability, experience, and attentiveness of the cytopathologists are critical to accurate lesion recognition. Thus, there is a pressing need for new technologies to address objective recognition and image processing issues to assist in disease diagnosis. 

Artificial intelligence (AI) has recently been slowly adopted in the medical field due to its strong self-learning ability and unbiased nature. Though the use of AI for SPLs is constrained and still developing in comparison to other fields, EUS-FNA/B has shown promising potential [27,28,29]. In this paper, we provide a comprehensive review to elucidate the progress and current prospects of EUS-FNA/B with AI for the diagnosis and differential diagnosis of SPLs.

## 2. Definitions of Artificial Intelligence, Machine Learning, and Deep Learning

Several publications have discussed AI, machine learning (ML), and deep learning (DL); yet, confusion around the terminology still exists. These terms are highly pertinent and cannot be used interchangeably (Figure 1). As a result, we aim to explain these concepts to the clinical audience in an accessible manner that avoids technical jargon.

AI is the ability of digital computers or computer-controlled robots to interpret information or perform tasks commonly associated with human intelligence [30]. It can be classified into three categories: artificial narrow intelligence (ANI, also known as weak AI), artificial general intelligence (AGI, also known as strong AI), and artificial superintelligence (ASI) [31]. ANI is goal-oriented and utilized to perform particular or limited tasks. Almost all AI systems, including medical AI systems, belong to the ANI category [32]. In contrast, AGI remains a theoretical concept that has not yet been achieved. An AGI system would consist of thousands of ANI systems and possess human-level cognitive function, allowing it to solve problems without human intervention [33]. Furthermore, ASI holds the potential to surpass human civilization and is the ultimate goal of AI creation [34]. However, it is unlikely to become a reality within the foreseeable future.

ML is a branch of AI that involves the use of algorithms to extract features from available data in order to make accurate predictions [35]. The demand for biomedical images and automatic analysis has led to significant advances in ML over the past decade, with a wide range of techniques, such as support vector machines (SVMs), random forests (RFs), decision trees, logistic regression analysis, and neural networks being employed [36]. DL is a neural network architecture that has evolved from ML and is characterized by a large number of interconnected elements that can automatically extract features from data, akin to the functioning of the human brain [36]. Convolutional neural networks (CNNs) are a common DL method primarily used to handle data with a grid-like topology, such as images (2D grid of pixels) or videos (3D grid of pixels). Initially, the CNN model is trained using a large collection of labeled images. Composed of multiple convolutional layers, activation functions, and pooling layers, the CNN automatically extracts features from the data [37]. Once trained, it can quickly and efficiently analyze new input images. CNNs have shown exceptional performance in analyzing and classifying medical images. In specific tasks, such as the detection of skin cancer or the identification of PDAC, CNNs trained on annotated datasets have been shown to exceed the accuracy of human experts [38,39,40,41].

## 3. Use of Artificial Intelligence in EUS-FNA/B

As computational power continues to increase and clinical demand grows, there have been significant advances in the utilization of AI to interpret complex images, particularly in EUS-FNA/B. In the field of pancreatic EUS-FNA/B, AI is predominantly used to aid in pathological diagnosis and is used less frequently in real-time puncture site guidance. The following sections will delve into the topics mentioned above in greater detail.

### 3.1. AI and Digital Pathology

Pathological images are a crucial form of biomedical imagery used for clinical pathological diagnoses, offering intuitive and valuable insights. Microscopic examination of these images is considered the gold standard for accurately determining the nature and presence of diseases during the diagnostic process [42]. Historically, the analysis of pancreatic specimens obtained via FNA or FNB has been the domain of professional pathologists. However, achieving precise pathological diagnoses and classifications is time- and labor-intensive, requiring pathologists to identify cellular and tissue characteristics and patterns indicative of pathological changes. Although training and standard guidelines can facilitate the harmonization of analytical processes, the subjectivity of pathological analysis and differences in visual perception, data integration, and judgment among independent observers inherently limit its reliability [43]. As a result, even pathologists with equivalent training may encounter diagnostic inconsistencies and discrepancies in opinion.

Digital pathology (DP) is the process of digitizing pathology information, including its acquisition, management, sharing, and interpretation, in a digital environment [44]. This technology enables the transformation of glass slides into digital ones that can be viewed on a computer monitor, offering two main benefits: improved efficiency and productivity and the integration of computer-aided diagnostic techniques [45]. With DP, team annotation of slides is possible, providing pathologists with greater flexibility in work schedules and remote access to pathology data. This technology also facilitates faster consultation telepathology turnaround times; delivers immediate access to previously archived digital slides; and streamlines data retrieval, matching, and organization [46]. Moreover, digital pathology algorithms enable the automatic quantification and analysis of pathology data, providing greater consistency and diagnostic accuracy than light microscopy and glass slides [47]. Given these advantages, DP is seeing increasing use for diagnostic, educational, and research purposes and is on the verge of becoming a mainstream option for routine diagnostics [48].

The potential for AI development in supporting pathology diagnosis, particularly in image analysis and disease detection, is significant. When applied to DP, AI algorithms can enhance the accuracy and reproducibility of morphological variables that pathologists traditionally assess. These algorithms can mine image features from DP slides, including visible morphology and spatial features, such as nuclear and gland size, shape, and tissue architecture. Furthermore, AI can extract features that pathologists may not recognize, such as intensity, texture, and spectral features [49]. These complex features can then be utilized to train models and perform specific segmentation, diagnostic, or prognostic tasks.

Feature extraction in AI involves two general approaches: supervised learning and unsupervised learning [50]. In supervised learning, features are identified based on the regions of interest (ROIs) annotated by pathologists in the images. These identified ROIs can be linked to specific, measurable attributes in the image and have some degree of explainability. Conversely, unsupervised learning uses algorithms to recognize patterns and similarities in image properties among training exemplars. These patterns may coincide with existing morphological classifiers; but, in some cases, they may be unknown to pathologists. To ensure the reliable diagnosis performance of AI algorithms in the medical field, it is crucial to evaluate their diagnostic reliability rigorously. Interpretability methods, such as Grad-CAM and AGF-Visualization, generate visual explanations for corresponding class labels, increasing the transparency of AI algorithms and enabling human scrutiny to detect undesirable AI behavior (Figure 2) [51,52]. The combination of AI and DP can determine each case’s objective measurement criteria and metrics, renewing pathologists’ interest in AI evaluation. This technology has already been authorized for clinical practice use in some geographical areas [53].

### 3.2. AI in Assisting with Pathological Diagnosis

Recent domestic and international studies demonstrate that AI’s diagnostic accuracy for DP images is comparable to that of senior pathologists, providing faster, more accurate, efficient, and collaborative pathological diagnoses [39,54,55]. AI can be particularly valuable in supporting clinical pathological diagnoses when pathologists are unavailable. As early as 1998, AI was reported to assist in diagnosing DP slides, marking a starting point in the pursuit of computer-aided early diagnosis of PC [56]. Currently, limited research integrating AI with pancreatic pathological diagnoses focuses on extracting nuclear features related to DNA content and chromatin distribution from ERCP cytological specimens and surgically resected histological specimens [57,58,59]. For instance, Song et al. developed and assessed an SVM model for automatically diagnosing and grading PDAC based on the morphologic features found on histology slides, achieving an accuracy of 94.38% in binary classification between PDAC and normal tissues [60]. This outcome suggests a tremendous potential for this model as a valuable supplement for the morphological evaluation of tumor biological characteristics. However, there remains a significant gap in the integration of AI-assisted diagnosis of DP images through specimens obtained via EUS-guided sampling.

Table 1 summarizes published studies that have used artificial intelligence to analyze DP images of EUS-FNA/B data, particularly those of SPLs. In 2017, Momeni-Boroujeni et al. reported the use of a multilayer perceptron neural network (MNN) in classifying pancreatic specimens obtained using EUS-FNA as benign or malignant, which was the first study available for cytological analysis using FNA/FNB samples [61]. The process involved using a K-means clustering algorithm to segment cell cluster pictures collected from FNA and extracting their morphological features. The MNN was then trained using differences in significant morphological features between malignant and benign images, such as contour, perimeter, and area. The MNN was successfully tested with a 100% accuracy rate in discriminating between benign and malignant pancreatic cytology while 77% accuracy was achieved for the atypical dataset. Additionally, a few original research papers and draft conference abstracts on the pathological classification of solid pancreatic masses were published. These papers used a small sample of cytopathological slides obtained through EUS-FNA, which had a limited diagnostic performance in single-center validation (accuracy range: 80–94%) [54,55,62]. Hyperspectral imaging (HSI) is a new optical diagnostic technology that combines spectroscopy. It measures the interaction between tissues and light through an HSI camera, capturing spectral features that conventional imaging modalities cannot obtain [63]. In this way, HSI can provide more diagnostic information for identification and differentiation. Qin et al. developed a CNN model combined with HSI technology, which used informative spectral features to distinguish benign and malignant pancreatic cytology [64]. By comparing the AI model’s diagnostic performance regarding the HSI images to conventional RGB images, one thing that can be learned is that the spectral information makes the CNN model easier to use to identify PDAC cells in cytological slides (HSI accuracy, 88.05%; RGB accuracy, 82.47%). Finally, the HSI-based model has been proven to have good generalization ability (internal test dataset: accuracy, 92.04%; external test dataset: accuracy, 92.27%). In 2022, Zhang et al. conducted a prospective, retrospective study using a novel deep CNN (DCNN) system to segment stained cell clusters and identify PC in a ROSE during EUS-FNA [39]. This study is the first known and the largest one to establish a deep learning system for identifying PDAC in a ROSE, including 6667 images from 194 cases and achieving an accuracy of 94.4% on the internal testing dataset. Additionally, the DCNN system demonstrated outstanding generalization ability on external testing datasets, with an accuracy of 91.2–95.8%. Moreover, its accuracy was comparable to cytopathologists and exhibited high sensitivity and negative predictive value (NPV). These results suggest that deploying the DCNN system in clinical settings to produce a ROSE may increase the diagnostic yield of EUS-FNA. In the same year, Lin et al. reported on a ROSE-AI model that substitutes manual ROSE during EUS-FNA [65]. It performed well in detecting cancer cells, presenting an 83.4% accuracy rate in the internal validation dataset and a similar result in the external validation dataset (88.7%). The ROSE-AI model’s implementation can speed up slide evaluation and shorten endoscopists’ wait times. Although AI has achieved promising results in ROSE, prospective validation studies are necessary to provide high-level evidence in actual clinical practice. We believe that future AI strategies will alleviate the problem of insufficient pathological resources and aid endoscopists in performing ROSE, thereby improving the accuracy of pancreatic disease diagnoses.

The development of FNB needles has made it possible to collect bigger tissue samples with fewer needle passes. With the introduction of EUS-FNB, several researchers contend that ROSE may no longer be required to minimize the number of needle passes [71,72]. In 2021, Naito et al. developed a CNN model for evaluating PDAC in EUS-FNB whole slide images (WSI), achieving a high ROC-AUC of 0.984 and an accuracy of 94.17% [68]. This model can assist in obtaining accurate histopathological diagnoses while avoiding interference from high blood, inflammatory, and digestive tract cell levels. However, a global survey of ROSE indicates that only 50% of Asian endoscopy centers meet the qualification standards for this technique [73]. In such cases, researchers suggest that a more fair and replicable evaluation method should be developed. Macroscopic on-site evaluation (MOSE) refers to the visual assessment of samples obtained during EUS-FNA/B and can serve as an alternative to ROSE [74]. Ordinarily, tumor tissues are white or flesh-colored while blood clots are red. During the period of MOSE, endoscopists transfer the puncture samples onto a glass slide and make a preliminary separation to observe the length of the white tissue samples, thereby assessing the adequacy of the aspiration or biopsy. This step provides an important basis for ensuring diagnostic accuracy. Nonetheless, evaluating specimen adequacy currently relies on the endoscopist’s subjective judgment, which largely depends on their level of experience. As such, it is crucial to develop more objective and reproducible evaluation methods for MOSE. However, in 2022, Ishikawa et al. reported on a contrastive learning-based CNN model that has achieved a comparable accuracy rate (84.4%) to endoscopists in evaluating the diagnosability of EUS-FNB specimens in MOSE [70]. This suggests that in the future, novel AI-based evaluation methods will replace MOSE, resulting in significant time savings and increased productivity.

### 3.3. AI in Guiding Targeted EUS-FNA/B

Although AI has made significant strides in pathological diagnosis, few reports have investigated its potential in lesion recognition and localization during EUS-FNA/B. CH-EUS is a cutting-edge technology that uses microbubble contrast agents to visualize microvessels and parenchymal perfusion, resulting in better characterization of pancreatic lesions detected by EUS [75]. Compared to conventional EUS, CH-EUS enhances the observation of pancreatic tumors and assists in identifying various pathological areas within pancreatic lesions. Combining CH-EUS with EUS-FNA notably captures subtle lesions that are not distinguishable from conventional EUS, thereby avoiding the sampling of necrotic areas and reducing the need for additional needle passes [76,77]. In a retrospective study comparing diagnostic accuracy and sampling adequacy between CH-EUS-FNA and conventional EUS-FNA groups, biopsy specimens were more frequently obtained in the CH-EUS-FNA group (96.6%) than in the EUS-FNA group (86.7%), with no significant difference in diagnostic accuracy [78]. Additionally, TIC has been used to achieve objective quantitative analyses of SPLs during CH-EUS, which includes variables such as maximum intensity gain, echo intensity reduction rate, and time to peak, enabling SPL classification based on enhancement patterns [79,80,81]. In 2023, Tang et al. developed an innovative auxiliary diagnosis system (CH-EUS MASTER) that uses AI models to guide targeted EUS-FNA/B procedures, which is the first time this technology has been utilized in this way [82]. By employing DCNN and RF algorithms, CH-EUS MASTER has achieved three crucial functions, including real-time pancreatic mass capture and segmentation under CH-EUS, identification of benign and malignant pancreatic masses according to TIC characteristics, and identifying and providing guidance for the target area of EUS-FNA. Endoscopists can perform further puncture procedures based on the ROIs predicted by CH-EUS MASTER with a remarkable accuracy rate of 93.8%, a sensitivity rate of 90.9%, and a specificity rate of 100%. Significantly, CH-EUS MASTER-guided EUS-FNA can improve the first-pass diagnostic yield (80.0% vs. 33.3%) compared to traditional EUS-FNA. Therefore, AI has the potential to assist in the pathological diagnosis of EUS-FNA/B and play a crucial role in guiding puncture sites, allowing inexperienced endoscopists to shorten their learning cycles. Although AI use in guiding targeted EUS-FNA/B is a relatively new field, future research in this area could produce innovative advancements.

## 4. The Limitations and Shortages of Artificial Intelligence in EUS-FNA/B

Although AI models are still in their infancy, they have already proven to be quite useful in organizing patient treatment procedures and assisting with medical decision making. However, many challenges remain to be addressed, particularly in terms of achieving an accurate diagnosis of specimens in EUS-FNA/B. Like any diagnostic tool, AI-assisted diagnostic models have their own set of limitations and shortcomings that need to be overcome before they can be considered reliable diagnostic methods for SPLs.

Building confidence in AI-assisted diagnostic models as a valuable tool in modern medicine requires addressing one of the most significant limitations, known as the “opaqueness” of AI, where the reasoning and recognition of the computer are not visible, leading to the “black box problem” [83]. This phenomenon can result in misdiagnosis without a clear understanding of why a particular decision was made, creating a fatal flaw in evidence-driven medicine. One potential solution to this issue is using inherently interpretable models that allow visualization of the regions recognized by AI as being important [84]. Another suggestion is conducting meticulous quality assessments before implementing AI models in the clinic to prevent physicians from relying solely on AI models to evaluate clinical outcomes [85]. This raises intricate regulatory and ethical considerations. AI systems designed for medical applications must be subjected to rigorous scrutiny and long-term validation to secure official certification.

Another major concern is the need for more standardization of the input data used to train AI models [86]. Establishing uniform data collection, processing, storage, reproduction, and analysis protocols is essential to ensure consistency. Without standardized data, the same AI model could produce vastly different outcomes for the same patients, reinforcing bias and leading to poor prognoses, which may reduce the popularity of AI technology. Additionally, an AI model trained in a specific environment may not perform equally well in different environments or on different devices. For instance, a deep learning CNN that is trained to accurately classify pancreatic biopsies stained with hematoxylin and eosin (H&E) may perform poorly or not at all on pancreatic biopsies prepared and stained with Papanicolaou (PAP). Other factors that can affect standardization settings include staining quality interference. Creating pathological smears requires experienced cytotechnologists to ensure that smears are uniform in thickness, well concentrated, and easily distributed, facilitating observation. Moreover, during staining, operators should ensure that the cells are fully coated with dye solutions to avoid blurry staining results. Untimely drying after staining can also lead to uneven staining, thus affecting image quality. Although it will be laborious and expensive to create uniform protocols for input data, they are necessary to improve the generalization of AI models.

Other factors that limit the development of AI in the field of EUS-FNA/B include the inability to fully utilize image information and the higher costs of image annotation. In particular, the algorithm’s operation relies on the graphics processing unit (GPU); but, the current storage capacity of GPUs is limited, making it challenging to fully utilize all the information in whole slide images (WSIs) or other image formats, which can result in the loss of some useful information [87,88]. Additionally, supervised learning is a common approach for most CNNs used in deep learning, which requires pathologists to accurately label ROIs in the images, adding to the cost [89]. Furthermore, AI analysis relies on high-quality training datasets, which require a substantial number of training images and can be time-consuming to prepare [90].

Regarding the application of AI in guiding puncture sites during EUS-FNA/B, several limitations and shortcomings need to be considered beyond the previously discussed data standardization issue. One critical limitation is that AI is challenging for dynamic image recognition [91]. EUS images are susceptible to external elements that can cause image jitter and displacement, such as a patient’s breathing and heartbeat. AI models must perform real-time corrections and registrations of EUS images to compensate for these discrepancies. 

Currently, there is a growing desire to use artificial intelligence as an alternative to tissue sampling, thereby eliminating the need for and adverse events associated with the procedure. As such, there is increasing interest in using AI-assisted EUS for the diagnosis of pancreatic lesions, mainly due to its relatively low cost and minimal invasiveness. Although an increasing body of research supports the superiority of AI-assisted EUS in diagnostic accuracy compared to traditional human interpretation, most clinicians remain cautious about its widespread application in clinical practice [86]. However, with ongoing improvements in AI algorithms and the quality of EUS images, AI-assisted EUS models have the potential to replace traditional EUS-FNA/B as the gold standard for diagnosing SPLs.

Despite the progress made by AI in the field of digestive endoscopy, its application in actual clinical practice has been limited by insufficient medical data and the need for high accuracy. Figure 3 provides an overview of the limitations linked to AI in EUS-FNA/B. To accelerate the utilization of AI for clinical diagnosis and treatment, it is essential to conduct prospective and multicenter research studies encompassing a wide range of medical images for AI model processing and analysis [92]. Such an approach would ensure the representativeness of the collected data and enhance the recognition of diagnostic results in the medical community.

## 5. Conclusions and Prospect

In summary, AI has been widely utilized in the EUS-FNA/B field, significantly advancing automatic pathological image diagnosis. Some AI models demonstrate diagnostic accuracy comparable to experienced pathologists. Further AI-assisted interpretation will mitigate empirical misjudgments among pathologists, thereby improving work efficiency and credibility. AI technology can also substitute for cytopathologists in ROSE and alleviate resource constraints. AI can assess the diagnosable rate of puncture specimens by MOSE, replacing endoscopists and saving operational time. Additionally, AI can identify the puncture site’s nature and guide target puncture areas during EUS-FNA/B to shorten endoscopist growth cycles. Nevertheless, current clinical AI applications in our field and country remain limited. Thus, future studies should focus on multicenter research to analyze and process data, facilitate AI’s clinical promotion, and make clinical diagnosis and treatment more feasible.

## Figures and Tables

**Figure 1 diagnostics-13-03054-f001:**
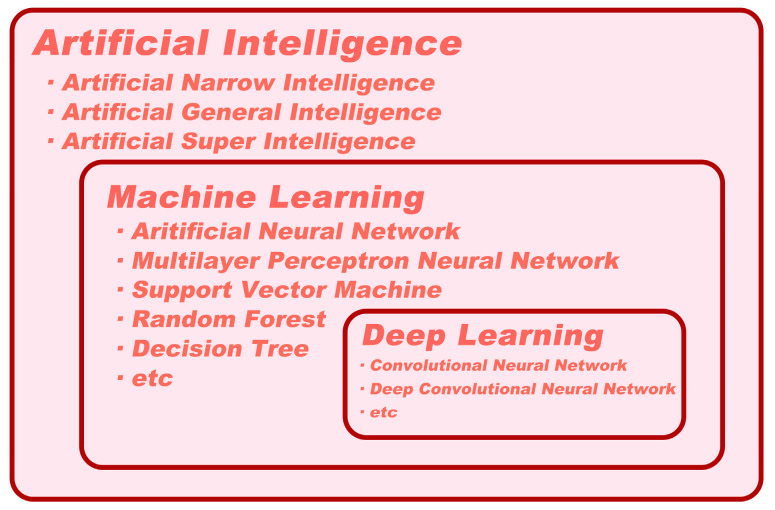
Types of artificial intelligence.

**Figure 2 diagnostics-13-03054-f002:**
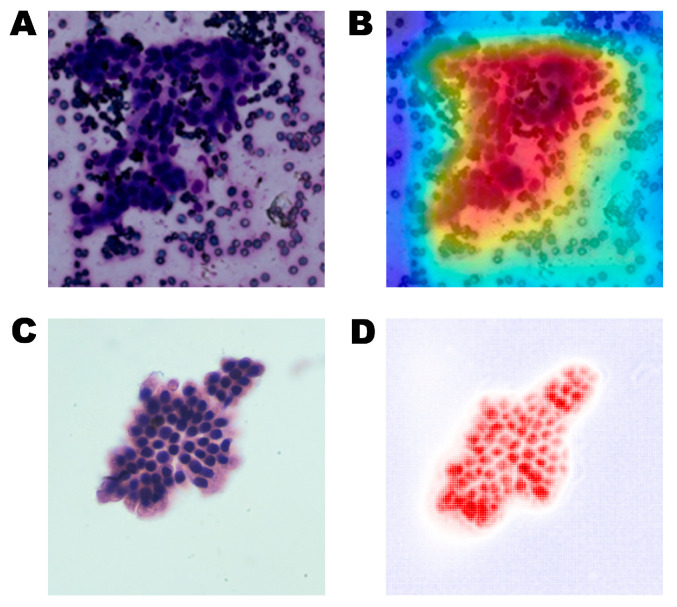
Commonly used interpretability methods to visualize pathological images. (**A**) an original pathological image; (**B**) a class activation map using the Grad-CAM for the pathological image; (**C**) the other original pathological image, and (**D**) a class activation map using AGF-Visualization. The high-intensity area (red color) reflects the area of interest to the AI model.

**Figure 3 diagnostics-13-03054-f003:**
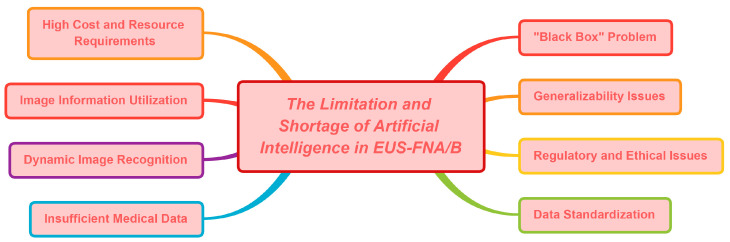
The limitations and shortages of artificial intelligence in EUS-FNA/B.

**Table 1 diagnostics-13-03054-t001:** Application of AI in EUS-FNA/B for the pathological diagnosis of solid pancreatic lesions.

Year/Journal	Author	Ref.	Purpose	Data Source	Sample Size	Algorithm	Diagnostic Performance
2017/ Cancer Cytopathology	Momeni-Boroujeni et al.	[61]	Distinguish benign and malignant pancreatic cytology	EUS-FNA	277 images from 75 pancreatic FNA cases	MNN	**For benign and malignant categories**: Accuracy 100% **For atypical cases**: Accuracy 77%
2018/ Gastrointestinal Endoscopy	Hashimoto et al.	[62]	PDAC identification	EUS-FNA	450 images	CNN	Accuracy 80%
2019/ Endoscopic Ultrasound	Kong et al.	[66]	PC detection	EUS-FNA	142 cases	DIA	Accuracy (83%) is comparable to conventional cytology (78%)
2020/ Gastroenterology	Hashimoto et al.	[54]	Distinguish benign and malignant in ROSE	EUS-FNA	**Retrospectively collected**: 1440 cytology specimens; **Retrospective validated**: 400 cytology specimens	CNN	Accuracy (93–94%) is comparable to an onsite pathologist (98–99%)
2021/ Gastroenterology	Thosani et al.	[67]	Interpretation for adequacy and identification of SPLs in ROSE	EUS-FNA	400 cases for training and 77 images for validation	ML	**For onsite adequacy testing**: Accuracy 87.25%; **For cytopathological diagnosis**: Accuracy 81.8%
2021/ Gastrointestinal Endoscopy	Patel et al.	[55]	Comparison of AI and subspecialty physicians for identification of SPLs	EUS-FNA	77 images	ML	Accuracy (87%) is on par or superior compared to most physicians (36–96%)
2021/ Scientific Reports	Naito et al.	[68]	PDAC detection in WSIs	EUS-FNB	532 WSIs	CNN	Accuracy 94.17%, AUC 0.9836
2022/ Diagnostics (Basel)	Yamada et al.	[69]	Distinguish PDAC and benign pancreatic cytology	EUS-FNA/B	246 specimens	DL	Accuracy 74%
2022/ Diagnostics (Basel)	Ishikawa et al.	[70]	Evaluation of diagnosable EUS-FNB specimen in MOSE	EUS-FNB	271 specimens from 159 patients	CNN	Accuracy (84.4%) is comparable to endoscopists (82.1–83.2%)
2022/ EBioMedicine	Zhang et al.	[39]	Identification of PDAC in ROSE	EUS-FNA	6667 images from 194 cases	DCNN	Accuracy (94.4%) with AUC 0.958, is comparable to cytopathologists (91.7%)
2022/ Journal of Gastroenterology and Hepatology	Lin et al.	[65]	Detection of cancer cells with pancreatic or other celiac lesions in ROSE	EUS-FNA	1160 images from 51 cases	CNN	**For internal validation dataset**: Accuracy 83.4% **For external validation dataset**: Accuracy 88.7%
2023/ Cancer Medicine	Qin et al.	[64]	Distinguish benign and malignant masses via pancreatic cytology	EUS-FNA	1913 images from 72 cases	CNN	**For internal test dataset**: Accuracy 92.04% **For external test dataset**: Accuracy 92.27%

Abbreviations: PDAC, pancreatic ductal adenocarcinoma; MNN, multilayer perceptron neural network; ML, machine learning; DL, deep learning; WSI, whole slide image; AUC, area under the ROC curves; DCNN, deep convolutional neural network.

## Data Availability

The datasets analyzed during the present research are available from the corresponding author upon reasonable request.

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
