# Peer review of "Artificial Intelligence in Endoscopic Ultrasonography-Guided Fine-Needle Aspiration/Biopsy (EUS-FNA/B) for Solid Pancreatic Lesions: Opportunities and Challenges"

_diagnostics, 2023, doi:10.3390/diagnostics13193054_

Round 1
Reviewer 1 Report
The authors have reviewed the research on the use of artificial intelligence for EUS-guided tissue acquisition, summarizing it very well. However, I have several concerns which are listed below.
#1 On page 3, lines 113 to 116, the authors mentioned CNN, but it is difficult to understand this sentence. A specific description is desirable.
#2 From page 5, line 212, the authors described hyperspectral imaging (HSI) technology, please explain in detail.
#3 From page 2, line 73, ROSE was mentioned and from page 7, line 4, MOSE was mentioned. Specific details should be explained. Also, there was no description of the full form of the abbreviation MOSE.
#4 On page 5, line 212, the authors described hyperspectral imaging (HSI) technology, please explain in detail.
# On page 6, in table 1, in the Algorithm column, there is a mention of ML and deep learning. I have no idea what this means. Please describe it in more detail.
# In table 1 on page 6, the abbreviation for “Deep Learning” is DP. However, in the manuscript, the full form of DP was “Digital pathology”. Please change it as it is confusing.
Also, the full form of MNN is not mentioned.
#5 On page 3, line 105, AI and ML full forms were being re-submitted.
#6 On page 3, line 110, DL's full form was being re-submitted.
None
Author Response
#1 On page 3, lines 113 to 116, the authors mentioned CNN, but it is difficult to understand this sentence. A specific description is desirable.
Response: We thank the reviewer for pointing this out. We have added the relative content of CNN and provided the specific description. Specifically as follows: Convolutional Neural Networks (CNN) are a common DL method primarily used to handle data with a grid-like topology, such as images (2D grid of pixels) or videos (3D grid of pixels). Initially, CNN model is trained using a large collection of labeled images. Composed of multiple convolutional layers, activation functions, and pooling layers, the CNN automatically extracts features from the data [37]. Once trained, it can quickly and efficiently analyze new input images. CNN has shown exceptional performance in analyzing and classifying medical images. In specific tasks, such as the detection of skin cancer or the identification of PDAC, CNNs trained on annotated datasets have been shown to exceed the accuracy of human experts [38, 39, 40, 41].
#2 From page 5, line 212, the authors described hyperspectral imaging (HSI) technology, please explain in detail.
Response: Thanks for your kind comments. We added the related content of HSI technology in our manuscript. “Hyperspectral imaging (HSI) is a new optical diagnostic technology that combines spectroscopy. It measures the interaction between tissues and light through an HSI camera, capturing spectral features that conventional imaging modalities cannot obtain [63]. In this way, HSI can provide more diagnostic information for identification and differentiation. Qin et al. developed a CNN model combined with HSI technology, which used informative spectral features to distinguish benign and malignant pancreatic cytology”.
#3 From page 2, line 73, ROSE was mentioned and from page 7, line 4, MOSE was mentioned. Specific details should be explained. Also, there was no description of the full form of the abbreviation MOSE.
Response: We thank the reviewer for pointing this out, and we are glad to answer this question. In our revised-manuscript, we exlained ROSE and MOSE in detail. Revised portion are marked in the paper and we highlight the changes to our manuscript by using colored text. In page 2, line 90, we have added the related content of ROSE. In page 7, line 273, we have added the related content of MOSE.
#4 On page 5, line 212, the authors described hyperspectral imaging (HSI) technology, please explain in detail.
Response: We thank the reviewer for pointing this out. Question 4 is same as Question 2. We added the related content of HSI technology in our manuscript. “Hyperspectral imaging (HSI) is a new optical diagnostic technology that combines spectroscopy. It measures the interaction between tissues and light through an HSI camera, capturing spectral features that conventional imaging modalities cannot obtain [63]. In this way, HSI can provide more diagnostic information for identification and differentiation. Qin et al. developed a CNN model combined with HSI technology, which used informative spectral features to distinguish benign and malignant pancreatic cytology”.
# On page 6, in table 1, in the Algorithm column, there is a mention of ML and deep learning. I have no idea what this means. Please describe it in more detail.
Response: "We appreciate the reviewer for highlighting this. 'ML' stands for 'machine learning.' However, this question is indeed challenging to address. We've carefully reviewed the original text multiple times and gathered relevant information online, but we still can't determine the specific algorithm type.
# In table 1 on page 6, the abbreviation for “Deep Learning” is DP. However, in the manuscript, the full form of DP was “Digital pathology”. Please change it as it is confusing.
Also, the full form of MNN is not mentioned.
Response: We thank the reviewer for pointing this out. We have changed the abbreviation for “Deep Learning”, and 'DL' stands for ' Deep Learning'. The full form of MNN is multilayer perceptron neural network.
#5 On page 3, line 105, AI and ML full forms were being re-submitted.
Response: Thank you for the reviewer's comments. We have removed the repeatedly mentioned full forms.
#6 On page 3, line 110, DL's full form was being re-submitted.
Response: Thank you for the reviewer's comments. We have removed the repeatedly mentioned content.
Reviewer 2 Report
Very interesting and comprehensive review. THese my comments:
1) Some further images would improve the quality of the paper
2) The authors should comment on the potential role of AI as a substitute of tissue sampling (we want AI to avoid the need of tissue sampling.....)
3) The authors should comment on the current state of the art of EUS-FNB, citing the most important meta-analyses on the topic (PMID: 35124072 ; PMID: 31031330)
4) The authors should comment on the other ancillary techniques currently available, such as contrast-enhancement EUS (cite PMID: 33481633 and PMID: 34217751)
Author Response
Response 2: Please provide your response for Point 2. (in red)
1) Some further images would improve the quality of the paper
Response: Thanks for your nice suggestions. We provided Figure 3 ”The Limitation and shortage of artificial intelligence in EUS-FNA/B” in order to help readers better understand our Section 4“The Limitation and Shortage of Artificial Intelligence in EUS-FNA/B”.
2) The authors should comment on the potential role of AI as a substitute of tissue sampling (we want AI to avoid the need of tissue sampling.....)
Response: Thanks for your comment. We have added the relative content of the potential role of AI as a substitute of tissue sampling. From page 9, line 373, specifically as follows: Currently, there is a growing desire to use artificial intelligence as an alternative to tissue sampling, thereby eliminating the need and adverse events associated with the procedure. As such, there is increasing interest in AI-assisted EUS for the diagnosis of pancreatic lesions, mainly due to its relatively low cost and minimal invasiveness. Although an increasing body of research supports the superiority of AI-assisted EUS in diagnostic accuracy compared to traditional human interpretation, most clinicians remain cautious about its widespread application in clinical practice [82]. However, with ongoing improvements in AI algorithms and the quality of EUS images, AI-assisted EUS models have the potential to replace traditional EUS-FNA/B as the gold standard for diagnosing SPLs.
3) The authors should comment on the current state of the art of EUS-FNB, citing the most important meta-analyses on the topic (PMID: 35124072IF: 7.7 Q1 ; PMID: 31031330)
Response: Thanks for your nice suggestions. In our revised manuscript, we cited the most important meta-analyses on the topic (PMID: 35124072IF: 7.7 Q1 ; PMID: 31031330). And the revised content as follows (page 2, line 78): “However, these findings do not standardize the used needle sizes and locations of the lesions. After settling on a 22G needle size, the diagnostic accuracy of FNB has not been proven to be significantly better than FNA [22]. While a network meta-analysis suggests that Franseen and Fork-tip needles may offer superior performance in tissue sampling of pancreatic masses, the confidence level in estimates remains low [23]”.
4) The authors should comment on the other ancillary techniques currently available, such as contrast-enhancement EUS (cite PMID: 33481633IF: 3.9 Q2 and PMID: 34217751)
Response: Thanks for your kind comments. We have added the content of the other ancillary techniques (CH-EUS and EUS elastography). And the relative content as follows (page 2, line 53): “The use of contrast agents in EUS can provide valuable information on the microvas-culature and perfusion within organs of interest, and hypoenhancing masses have been proven to be indicators of malignant tumors [9]. According to a range of studies previously reported, CH-EUS shows a sensitivity range of 80% to 96% and a specificity range of 64% to 100% for SPLs, particularly in differentiating PDAC from other pan-creatic masses [10]. Utilizing CH-EUS as an adjunct to assist EUS-FNA demonstrates higher diagnostic sensitivity in the assessment of pancreatic masses compared to standard EUS-FNA(CH-EUS-FNA, 84.6%; EUS-FNA, 75.3%) [11]. As for EUS elas-tography, it serves as a valuable complement to tissue sampling, which is used to guide fine-needle punctures and aid in determining further clinical treatment plans [12]. Elastography has been reported to demonstrate an extremely high sensitivity (ranging from 92% to 98%) in the detection of malignant pancreatic tumors [13]. Specifically, it has an exceptionally high negative predictive value for diagnosing PDAC in small pancreatic lesions [14]”.
Round 2
Reviewer 2 Report
The revised version of the manuscript is OK. Thank you!
Author Response
/